# Effect of a Metal Conditioner on the Physicochemical Properties and Tribological Performance of the Engine Oil SAE 5W-30 API SN

Oriana Palma Calabokis [1,2,*], Yamid Nuñez de la Rosa [1,2], Vladimir Ballesteros-Ballesteros [2], Paulo César Borges [1] and Tiago Cousseau [1,3]

1 GrMatS Group, Universidade Tecnológica Federal do Paraná, Curitiba 81280-340, Brazil; yamid@alunos.utfpr.edu.br (Y.N.d.l.R.); pborges@utfpr.edu.br (P.C.B.)
2 Faculty of Engineering and Basic Sciences, Fundación Universitaria Los Libertadores, Bogotá 111221, Colombia
3 Centre for Bulk Solids and Particulate Technologies, The University of Newcastle, Shortland, NSW 2307, Australia
* Correspondence: calabokis@alunos.utfpr.edu.br

**Abstract:** Metal conditioners (MC) are added to lubricants to enhance their friction and wear in friction pairs, mainly in engines, gearboxes, and rolling bearings. Its growth in the Brazilian market is primarily focused on internal combustion engines. The effect of mixing MC with commercial engine oil (SAE 5W-30 API SN) was studied regarding the rheological and thermal properties. Also, the tribological performance of steel–steel contact was investigated. The rheological and thermal properties were determined by flow curves (at 20, 40, and 100 °C) and differential scanning calorimetry (DSC), respectively. Reciprocating fully-lubricated tests were performed at 40 °C and 80 °C (Po = 1.7 GPa, 5 Hz). Differences in the chemical composition between SAE 5W-30 and its mixture with MC were identified by infrared spectroscopy and related to their tribological performance. The coefficient of friction remained within the range of 0.09–0.1 for all conditions, typical of lubricated steel–steel contacts under boundary and mixed lubrication regimes. However, the mixture improved the wear resistance by around 33% when lubricated at 80 °C compared to the wear resistance offered by 5W-30. The formation of tribofilms with different chemical compositions was confirmed by SEM-EDS for all conditions. At both temperatures, the tribological performance reveals beneficial synergy between the metal conditioner and fully formulated oil additives. The tests lubricated with the mixture at 40 °C showed a less severe wear mechanism when compared to the tests lubricated with neat 5W-30. The study demonstrated that the mixture maintained the physicochemical properties of the commercial oil with a substantial anti-wear action at 80 °C.

**Keywords:** metal conditioner; tribology; friction; wear; internal combustion engine





## 1. Introduction

Currently, friction pairs are designed to perform in increasingly severe operating conditions without compromising energy efficiency. As a result, components inevitably operate between mixed and boundary lubrication regimes [1,2]. In such situations, the load is supported by the lubricant and the contact points between the surface's asperities [3]. Under specific combinations of load, speed, and temperature, a tribofilm may form through physical or chemical interactions between the lubricant and the contacting surfaces [4]. The presence of the tribofilm can affect friction and wear during tribological contact. Lubricating additives of extreme pressure (EP), anti-wear (AW), and friction modifiers (FMs) are utilized to facilitate the formation of this protective film [5]. Consequently, the combination of additives with different chemical compositions offers a potential solution for improving tribological performance across a wide range of operating conditions. However, as Eickworth et al. [5] discovered, interactions between additives can be synergistic, antagonistic,

or intermediate. Furthermore, Lyu et al. [1] confirmed that operating conditions such as temperature and load can influence the thickness and composition of the tribofilm, thereby affecting the tribological performance.

Due to the increasing demand for low-viscosity engine oils in recent decades, additives from the Zinc dialkyl dithiophosphate (ZDDP, ZDTP) family have been widely used [1,3,6]. Huynh et al. [6] found that mixtures of ZDDP with new additives (cyclopropane carboxylic acid and Ni nanoparticles) had both negative and positive effects on reducing wear in steel sliding contacts. Therefore, the development of new additive technologies must always be evaluated with fully formulated oils for validation [7,8]. Unfortunately, research related to the study of interactions between additives in fully formulated oils is less frequent due to their higher complexity. Pereira et al. [7] revealed that the tribofilms formed from engine oil (Mobil 1 5W-30) have different chemical composition, thickness, mechanical, and tribological properties compared to those formed by individual additives in base oils, as expected. Umer et al. [8] stated that the tribological performance during lubrication with 5W-30 engine oil is controlled by overcoming the energy barriers to activate the tribofilm-forming additives. Therefore, formulating new additives compatible with fully formulated oils is a permanent objective of the automotive industry.

Metal conditioners (MC) are lubricating products dominant in the Brazilian market that theoretically have a high chemical affinity for metallic surfaces. As a result, they form a protective film against wear and friction on components in tribological contact [9,10]. Information about their chemical composition can be found in American patents authored by Kusch [11], Stewart [12], and Roberts [13]. MC contains AW, FM, and EP-type additives, along with common oil additives such as detergents, dispersants, antioxidants, and viscosity improvers [11–13]. Although these products are highly commercialized, only two scientific papers have studied the effects of MCs on tribological performance. Coppini et al. [14] observed reductions in the wear of carbide drill inserts and increased service life and productivity due to the pre-treatment of drills with metal conditioner. In a previous publication, Calabokis et al. [15] found improvements in tribological performance in laboratory and field tests. The authors [15] tested the mixture of a metal conditioner with SAE 10W-30 API SL JASO MA oil on small-capacity motorcycles. However, the effects of adding these products on rheological, thermal, and thermo-oxidative properties still need to be investigated. Since the objective of MC is to reduce friction and wear, it is expected that their mixtures with engine oils do not affect the properties mentioned above that would impair lubrication, cooling, and degradation. Thus, this work evaluates the effect of adding a metal conditioner to commercial oil SAE 5W-30 API SN. The properties (rheological, thermal, and thermo-oxidative) and tribological performance were characterized and correlated with the chemical composition studied using infrared spectra.

## 2. Materials and Methods

The methodology was divided into two main aspects: Lubricant and tribological characterization.

### 2.1. Lubricant Characterization

The study focused on the effect of adding a metal conditioner to fully formulated oil for internal combustion engines. Table 1 provides the properties of the metal conditioner (MC) and engine oil based on their technical data sheets. The following lubricants were evaluated:

1.  Metal Conditioner (MC);
2.  Fully formulated commercial engine oil SAE 5W-30 API SN (5W-30);
3.  Mixture: Engine oil + Metal Conditioner in proportion 20:1—5% *v/v* of MC—(m5W-30): This mixture is recommended by the manufacturer of the MC for vehicle engines.

**Table 1.** Properties of lubricants according to data sheets.

| Technical Properties | Metal Conditioner | SAE 5W-30 API SN |
|---|---|---|
| Kinematic viscosity 40 °C (ASTM D445) (cSt) | 43.84 | 54 |
| Kinematic viscosity 80 °C (ASTM D445) (cSt) | 5.883 | 9.6 |
| Relative density (g/cm$^3$) | 1.098 | 0.84 |

Lubricant characterizations were performed using Fourier-transform infrared absorption spectroscopy (FTIR) under new conditions. A Varian spectrophotometer, model 640-IR, equipped with ATR (Attenuated total reflection) and a ZnSe crystal (Pike) was used. The measurement resolution was 4 cm$^{-1}$, and the reading range was 4000 to 650 cm$^{-1}$. Sixty-four scans were completed for each lubricant sample. The purpose of the FTIR measurements was to identify the characteristic peaks of the MC and determine if these peaks were present in the mixture, indicating the presence of the MC in lubricating oils. The spectrum bands and the organic groups were correlated using the methodology proposed by Lopes and Fascio [16] and compared with other references [3,17,18] and ASTM standards [19,20].

Differential Scanning Calorimetry (DSC) tests were conducted using a Perkin Elmer DSC4000 (Serial Number: 520A4063004). Two types of calorimetry analyses described in Table 2 were performed to investigate whether MC affects lubricant oxidation, thermal stability, and Oxidation Induction Time (OIT).

**Table 2.** Differential scanning calorimetry (DSC) test conditions.

| Condition | Description |
|---|---|
| **Non-isothermal scanning in different atmospheres [17,21]** | |
| Scan rate | 5 °C/min |
| Temperature | From room temperature (22 ± 5 °C) to 425 °C. |
| Gas flow and type | Flow rate: 50 mL/min: (1) Oxidizing atmosphere: synthetic air flow; (2) Inert atmosphere: nitrogen flow. |
| **Isothermal scanning in an oxidizing atmosphere according to ASTM E1858 [22]** | |
| Gas flow and type | Flow rate: 50 ± 2 mL/min//Oxidizing atmosphere: synthetic air flow |
| Heating cycle | Step 1: Heating rate of 40 °C/min from room temperature (22 ± 5 °C) to 195 ± 0.4 °C; Step 2: Maintain at 195 ± 0.4 °C (isothermal heating) and record the heat flux as a function of time; Step 3: Stop the heating when the exothermic oxidation peak is observed or until an inflection point is observed and the total displacement from the initial baseline exceeds 3 mW/g. Oxidation Induction Time (OIT) is the total time from the start of the experiment at room temperature in oxygen to the extrapolated onset time of the exothermic process. |

Rheological measurements were conducted to assess the potential viscosity changes that could occur with the addition of MC in the commercial lubricant. The rheological characterization of the lubricants involved obtaining flow curves at temperatures of 20 °C, 40 °C, and 100 °C. These curves illustrate the relationship between dynamic viscosity ($\eta$, Pa s) and shear rate ($\gamma$, s$^{-1}$). The measurements were performed using a TA Instruments Model DHR3 hybrid rheometer. Two different geometries were employed: (1) Parallel plates (PP, Diameter Ø: 40 mm, gap: 150 μm); (2) Couette geometry (CG, Bob Ø: 28 mm; Bob distance: 42.1 mm; reservoir Ø: 30.4 mm; gap: 2 mm). The imposed strain rate ranged

from $10^{-2}$ s$^{-1}$ to $10^3$ s$^{-1}$. Adequate repeatability was considered if consistent results were obtained with both geometries.

### 2.2. Tribological Characterization

Linear reciprocating tribological tests were carried out on a Bruker CETR-UMT-2MT s/n T1471 tribometer in the sphere-flat configuration with abundant lubrication (15 mL, 40 °C, and 80 °C). AISI 52100 steel spheres with a diameter of 10 mm were used as counter-bodies (Rq = 0.05 ± 0.01 μm, HRC = 60–66). The test specimens were low-alloy steel samples of AISI 4140 (HV$_{0.1}$ = 360 ± 30, dimensions: 20 × 10 mm$^2$, height: 8 mm). This steel was chosen because it is commonly used for manufacturing bearings, shafts, pins, and gears. The specimens were sanded to a surface finish between N4 and N5 (Rq = 0.4 μm) with sanding marks oriented perpendicular to the movement of the body, as suggested by Cousseau, Aceros, and Sinatora [23] to promote tribofilm formation. In addition to tests performed with 5W-30 and m5W-30, tribotests were performed with PAO 8 (polyalphaolefin 8, hereafter referred to as PAO) and its mixture with MC (mPAO, 20:1) to evaluate the overall effect of adding MC on the tribological performance of the base oil. PAO was selected because SAE 5W-30 API SN features a synthetic base oil type with a similar viscosity.

The reciprocating motion was maintained at a frequency of 5 Hz with a track length of 10 mm and a load of 50 N applied for 60 min. The coefficient of friction (CoF) was recorded at an acquisition frequency of 100 Hz. At least three samples of the core/counter body were tested for each lubricant. Detailed information on the selected operating conditions for the tribological tests can be found in the work of Calabokis et al. [15].

After the testing, the samples were cleaned in an ultrasonic bath with pure hexane. The worn surfaces were then characterized using scanning electron microscopy with energy-dispersive X-ray spectroscopy (SEM-EDS, Zeiss brand, model EVO MA 15). The objective of this characterization was to evaluate the wear mechanisms and provide a semi-quantitative chemical analysis of the tribofilms.

The wear volume was determined using optical profilometry with a green light interferometer (Talysurf Hobson, model Talysurf CCI Lite M12-3993-03, 20×) on the worn track. Firstly, surface leveling was performed, excluding the wear track. Subsequently, spurious points were eliminated (0.01–99.99%), and the surface data (matrix x, y, and z) was exported. The wear volume was computed using a MATLAB® code (Academic License Number 31568410) developed by the authors. This code determines the mean plane of the surface, excluding the wear zone, and then sums all the height values of the matrix, as advised in [24]. It also allows for the reconstruction of the 3D wear volume from the surface data. The results are presented as the mean and standard deviation of the measurements. Additionally, the specific lubricant film thickness (λ) was calculated according to Cousseau, Aceros, and Sinatora [23]. For this purpose, the topography regarding the Sq roughness parameter (Root Mean Square Height) was measured before and after the tribological tests using the TalyMap software (Platinum 6.1).

## 3. Results

### 3.1. Thermal Properties

Thermal and thermo-oxidative properties were determined using differential scanning calorimetry. The first methodology involved non-isothermal scans in both an inert (nitrogen) and an oxidizing atmosphere (synthetic air). Figure 1a shows the curves obtained in both atmospheres. When scanning in a synthetic air atmosphere, all the analyzed lubricants exhibited an oxidation degradation peak within the temperature range 22–425 °C. The mixture displayed a thermo-oxidative degradation peak at the same temperature (315 °C) as the 5W-30 oil. However, the oxidation peak of the MC was observed at 334 °C.

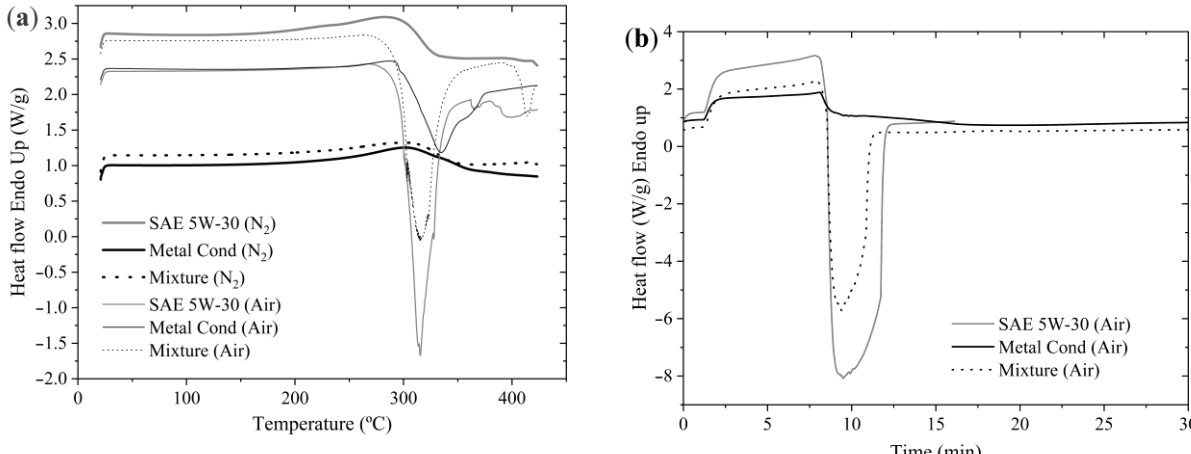

**Figure 1.** (**a**) Non-isothermal scanning curves in oxidizing and inert atmospheres (5 °C/min, 50 mL/min, 22–425 °C): synthetic air and nitrogen. (**b**) Isotherm curves for determining the oxidative induction time (OIT) according to ASTM E1858 (Synthetic air 50 mL/min, isotherm at 300 °C).

Figure 1a also indicates that none of the evaluated lubricants displayed a thermal degradation peak during scanning in an inert atmosphere. Thermal degradation peaks occur at much higher temperatures in inert atmospheres. Knowledge of the thermal degradation temperature is less relevant since the lubricant comes into contact with oxygen in internal combustion engines. Furthermore, the lifetime of the lubricant depends on its oxidation stability [3,21].

Figure 1b presents scan curves of the oxidative induction time (OIT). According to the ASTM E1858 standard, the test should be conducted in an oxygen atmosphere. However, due to the unavailability of this gas, it was substituted with synthetic air. This difference in atmosphere likely caused the exothermic peak not to occur at the temperature suggested by the standard (195 °C). As a result, several attempts were made at higher temperatures (250 °C, 280 °C, 295 °C, and 300 °C) to determine the OIT. Figure 1b shows that the lubricant 5W-30 and its mixture exhibited an exothermic degradation peak, indicating an OIT of 8.6 min. The MC did not show a degradation peak under the evaluated conditions, suggesting its superior thermo-oxidative stability, as confirmed by the non-isothermal scans (Figure 1a). Therefore, to determine the OIT of the MC, the procedure should be repeated at even higher temperatures (>300 °C). Finally, it was confirmed that the mixture maintains the same OIT as the SAE 5W-30 API SN oil.

The calorimetry results suggest the range of operating temperatures within which thermo-oxidative stability is maintained. Knowledge of the OIT and the temperature of thermo-oxidative degradation is essential for studying the performance of modern lubricants against excessive oxidation. The results revealed that the metal conditioner does not affect the thermal properties of the commercial lubricant SAE 5W-30 SN when used in the quantities suggested by the manufacturer (5% *v/v*). However, it is important to be aware that studying the oxidative and thermal stability of lubricants using conventional thermo-analytical techniques (TGA, DSC, and DTA) is not representative of real operating conditions due to factors such as the lack of moisture and metallic wear particles, combustion gases, fuel contamination, and service time, among others [21]. Nevertheless, a field test conducted on low-displacement motorcycles, previously developed by Calabokis et al. [15], confirmed the DSC results obtained in this study. The degradation of the crankcase lubricant remained within acceptable limits when a metal conditioner was added [15].

### 3.2. Functional Group Analysis

Fourier-transform infrared (FTIR) spectra were acquired from lubricants in their new, undegraded condition, as shown in Figure 2. Table 3 presents the most prominent

absorption bands and the corresponding organic functional groups associated with these characteristic bands.

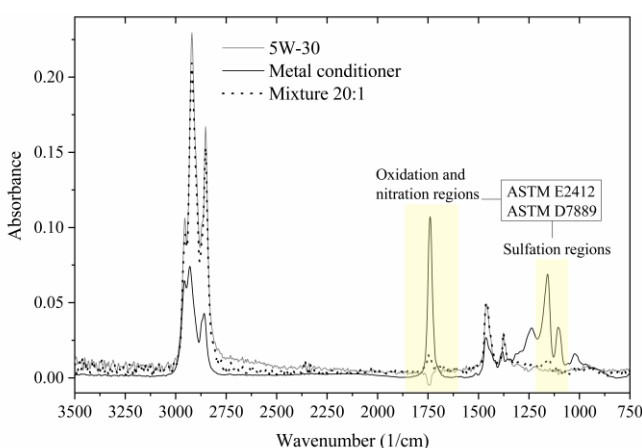

**Figure 2.** Infrared spectra (FTIR-ATR, ZnSE crystal) of commercial lubricant SAE 5W-30 SN and its mixture with metal conditioner.

**Table 3.** Correlation between infrared spectrum bands and probable organic functional groups. The interpretation was based on the methodology proposed by Lópes and Fascio [16]. (MC: metal conditioner; All: indicates that it was exhibited in all samples (5W-30, MC, and m5W-30); ν: stretching.

| Band (cm$^{-1}$) | Sample | Probable Functional Groups |
|---|---|---|
| 1031 | MC | AW additives (based on phosphate, mainly ZDDP) [19,20]; Hydrocarbons with bonds (νC-O): alkyl ether [16] |
| 1110 | MC | AW additives (P-O bond) [18]; Hydrocarbons with bonds (νC-O): alkyl ether [16] |
| 1160 | MC, m5W-30 | AW additives (P-O bond) [18]; Hydrocarbons with bonds (νC-O): ether [16] |
| 1243 | MC | AW additives (P-O, P=O bond) [18]; Hydrocarbons with bonds (νC-O): alkyl ether [16] |
| 1376 | All | Alkane hydrocarbons νCsp$^3$-H: CH3 (δ-symmetric) [16,17]; alkyl halides [16] |
| 1463 | All | Alkane hydrocarbons νCsp$^3$-H: CH2 (δ-scissor) [16] |
| 1741 | MC, m5W-30 | Hydrocarbons with bonds (νC=O): esthers [16] |
| 2854 | All | Alkane hydrocarbons νCsp$^3$-H [16]; carboxylic acids [3] |
| 2921 | All | Alkane hydrocarbons νCsp$^3$-H [16]; carboxylic acids [3] |
| 2958 | All | Alkane hydrocarbons νCsp$^3$-H [16]; carboxylic acids [3] |

Figure 2 clearly illustrates several differences between the spectra of the 5W-30 lubricant and the metal conditioner, as outlined in Table 3. The spectrum of the mixture combines the characteristic bands of each component.

The results of the band correlation analysis (Table 3) suggest that the base oils of the 5W-30 lubricant and the metal conditioner are primarily composed of alkane hydrocarbons (C-H bonds). Additionally, both may contain compounds with carboxylic acids, common in detergents, FM, AW, and EP additives. The 5W-30 oil does not contain phosphorus-based AW additives, which also exhibit antioxidant properties [3]. Furthermore, it lacks ether-type bonds (νC-O), unlike the analyzed metal conditioner (observed at 1031, 1110, 1160, and 1243 cm$^{-1}$). Notably, the MC displays a prominent band at 1741 cm$^{-1}$, indicative of hydrocarbons with ester-type bonds (νC=O). This observation could be associated with additives such as friction modifiers or a percentage of ester-type base oil [3,4]. Ester oil is more probable since no reduction in friction was observed during tribological characterization (Section 3.4). Also, the MC may contain alkyl halides (halogen bonded to an alkyl radical group). The functional groups identified in the MC (Table 3) suggest the presence of EP, AW,

antioxidant, and detergent additives in its formulation. Furthermore, Figure 2 highlights the regions associated with oxidation, nitration, and sulfation for used lubricating oils (with service time), as per ASTM E2412 [19] and ASTM D7889 [20] standards. It is worth noting that the infrared spectrum of the MC (Figure 2) exhibits absorption bands in these regions even for fresh samples (non-degraded), thus preventing a direct correlation between the intensity of these bands and degradation, as previously discussed by Calabokis et al. [15] in their study on used engine oils.

### 3.3. Rheological Properties

Viscosity is a crucial property of lubricating oils, as it plays a significant role in the performance of lubrication systems. An increase in viscosity results in a higher film thickness, providing several advantages for EHD (elastohydrodinamic) contacts [25]. However, it also leads to increased energy losses due to pumping and overall drag losses, which are particularly significant in internal combustion engines. Conversely, a decrease in viscosity, when combined with an appropriate additive package, has emerged as a new trend aimed at ensuring high wear resistance while minimizing friction losses [3]. Therefore, when adding foreign products (such as MCs) to fully formulated lubricants, it is important to maintain the rheological behavior of the lubricating oil as recommended by the original equipment manufacturer (OEM). This prevents excessive drag losses while preserving the functionality of the additive package.

To study the effects of adding a metal conditioner to the commercial lubricant, flow curves were measured using a rheometer with both parallel plate (PP) and Couette geometries (CG). Figure 3a presents the viscosity results as a function of the shear rate in the CG geometry. Notably, there was excellent reproducibility between both geometries, despite the different gap sizes employed (150 µm vs. 2 mm). The use of different geometries aimed to determine whether the non-Newtonian behavior observed at low shear rates was due to geometry-dependent measurement errors or an intrinsic response of the lubricant. The results confirmed that it was indeed an inherent characteristic of the lubricant.

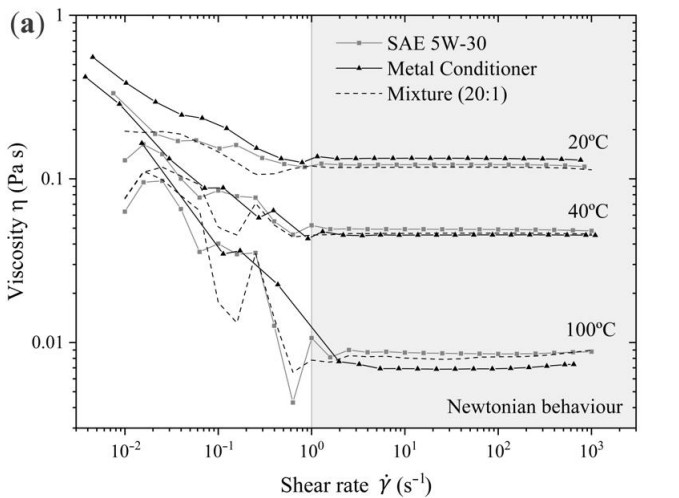
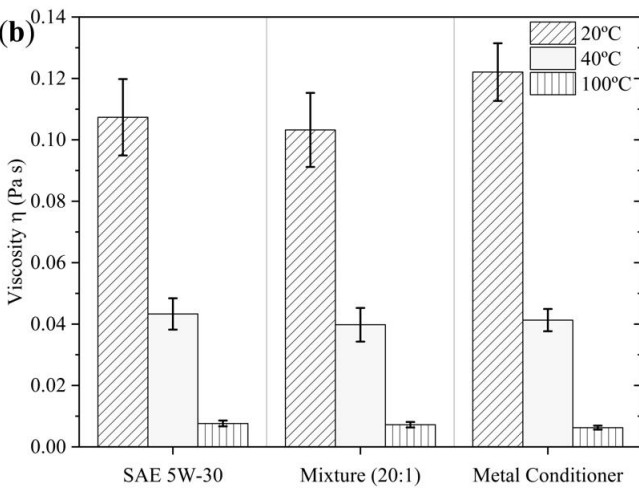

**Figure 3.** (**a**) Flow curves (Couette Geometry). (**b**) Dynamic viscosity as a function of temperature (Error bars represent standard deviation).

All evaluated lubricants demonstrated Newtonian behavior at strain rates > 1 s$^{-1}$ across all temperatures and geometries. However, in the region of low shear rates (<1 s$^{-1}$), all flow curves (Figure 3a) exhibited non-Newtonian behavior, specifically pseudoplastic (shear thinning behavior). It is worth noting that viscosity values for low shear rates are of lesser interest since the lubricated elements in the engine operate at very high shear rates [26–28]. Moreover, undesired outputs due to wall slip are common in the low shear

zone [26,29]. Therefore, the average of the viscosity values obtained from both geometries in the Newtonian regime was calculated, and the results are summarized in Figure 3b.

The literature reports the occurrence of non-Newtonian behavior in lubricating oils [26–28]. For instance, Taylor et al. [26] observed pseudoplastic behavior in the compression ring due to high contact pressures and shear rates. Similarly, the works of Moore et al. [27] and Meunier et al. [28] revealed that certain polymeric additives induce a rheological shift from Newtonian to pseudoplastic behavior at the onset of the EHD regime ($10^6$–$10^7$ s$^{-1}$) in automotive lubricants. In the present study (Figure 3a), the pseudoplastic behavior was observed at deformation rates < 1 s$^{-1}$, which is of limited relevance since the main engine and crankshaft elements operate at rates of $10^4$ s$^{-1}$. This shear thinning behavior (Figure 3a) may be associated with the findings of De Rosso and Negrão [29] regarding non-colloidal suspensions studied via parallel plate rheometry. According to the authors [29], non-Newtonian behaviors result from a combination of inertial and gravitational effects that affect the particle distribution on the rheometer plates, leading to reduced shear resistance. Another hypothesis relates to the alignment or disentanglement of long polymer chains during rheological measurements at low shear rates [27].

Finally, Figure 3 demonstrates that the mixture with metal conditioner (m5W-30) maintains the same rheological behavior and dynamic viscosity as the SAE 5W-30 oil at 20 °C, 40 °C, and 100 °C. Therefore, it does not change the lubricant flow within the engine components. This characteristic is crucial for proper lubrication, as increasing viscosity increases drag losses, while decreasing viscosity increases contact severity [30].

### 3.4. Tribological Characterization

The tribological response was evaluated in reciprocating sphere-on-plane tests, specifically in the boundary/mixed lubrication regime. The coefficient of friction (CoF) was measured, and the specific wear rate of the AISI 4140 steel body was calculated based on the quantification of wear track volume using 3D profilometry. Figure 4 presents the CoF and specific wear rate for all tested lubricants (PAO and 5W-30) as well as their mixtures with MC at a ratio of 20:1 (mPAO and m5W-30).

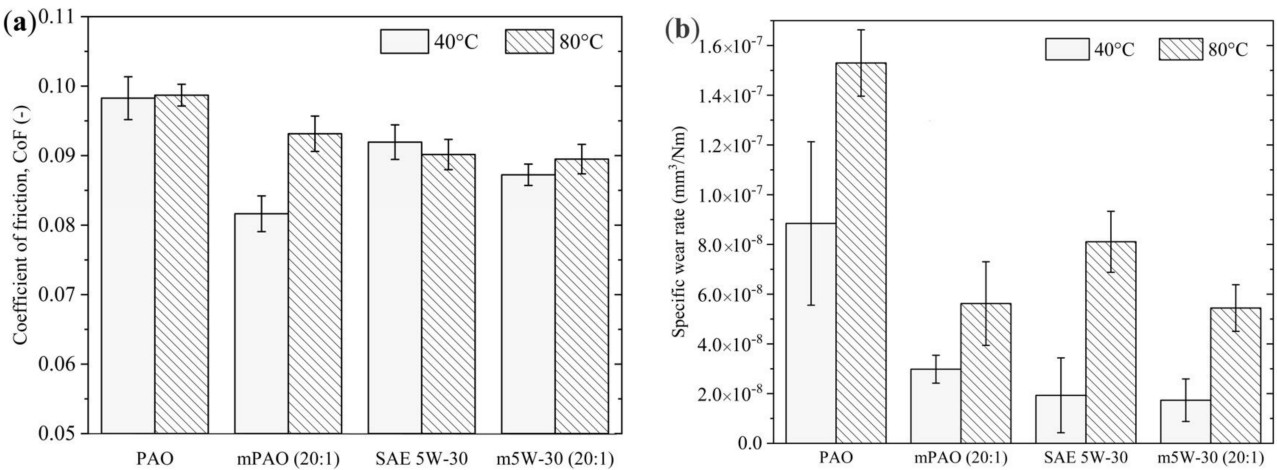

**Figure 4.** (**a**) Coefficient of friction (CoF). (**b**) Specific wear rate (mm$^3$/Nm) of the moving body. Reciprocating motion between the counter-body (AISI 52100, Ø10 mm) and moving-body (AISI 4140): $P_o$ = 1.7 GPa; 5 Hz; 10 mm stroke; 3600 s; abundant lubrication. Error bars represent the standard deviation. mPAO and m5W-30 represent the mixture of each oil with MC.

In Figure 4, the addition of just 5% of MC to the PAO base oil proved sufficient to reduce both the CoF and the wear rate to the range of values seen in the fully formulated 5W-30 oil. It is important to note that MC demonstrates its efficiency as an additive by maintaining an average wear rate of less than $6.0 \times 10^{-8}$ mm$^3$/Nm under the evaluated tribological conditions at both temperatures, regardless of whether it is added to PAO or

SAE 5W-30. It is worth highlighting that the application of MC in the automotive sector is limited to mixing it with fully formulated commercial oils. Therefore, there is no need for further investigation into the tribological study involving additive-free base oils, as it is irrelevant in this context.

The coefficient of friction (CoF, Figure 4a) remained at around 0.09 for both SAE 5W-30 and its mixture at both temperatures, which is typical for metal-to-metal lubricated contacts under early mixed and boundary lubrication [31]. The CoF as a function of time is displayed in Figure 5a. During the initial approximately 50 s of the test, a phenomenon known as the running-in period was observed. This transitional phase was characterized by a decrease in the CoF from its initial value. After the running-in period, the CoF stabilized at an approximately constant value (steady-state friction) for all conditions throughout the entire experiment.

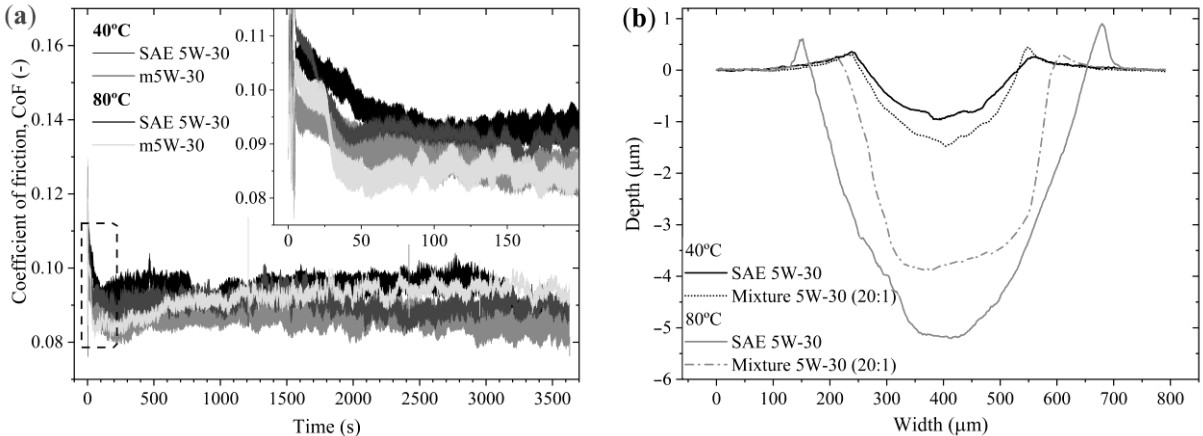

**Figure 5.** (**a**) Coefficient of friction as a function of time: an enlargement of the first 200 s is presented. (**b**) Depth profile of worn tracks.

Regarding wear performance, Figure 4b shows that at 40 °C, the wear rate was similar between the 5W-30 oil and the mixture. However, tests conducted at 80 °C revealed significant differences in the wear rate between the samples lubricated with SAE 5W-30 and m5W-30 (Figure 4b). These wear differences are further highlighted in the depth profiles of the wear tracks presented in Figure 5b. The cross-section profiles align with the wear rates depicted in Figure 5b: both profiles at 40 °C showed a similar track at width and depth. However, at 80 °C, the track lubricated with 5W-30 exhibited larger dimensions and a more pronounced pile-up compared to the track lubricated with m5W-30.

Representative SEM images were selected to examine the worn areas and infer the micro-wear mechanisms (Figure 6). Based on the SEM micrographs (Figure 6) and the tribological conditions imposed (boundary/mixed regime), it can be deduced that a tribochemical film was formed in all cases due to lubricated contact between the surfaces, as obtained in previous studies [1,5,32–35]. Within this regime, friction and wear control primarily occur through films generated by chemical and physical interactions between the additives and the surfaces of the moving bodies, known as tribofilms [4]. The tribofilms formed during the tribotests were identified and characterized using SEM-EDS. The semiquantitative chemical composition results are presented in Table 4.

The enlargements of Figure 6 reveal that the protective tribofilm experienced stress from repetitive contacts, leading to plastic deformation of the underlying metal surface under all tribological conditions. In the enlargements of Figure 6a–c, it is evident that specific areas underwent significant deformation with repeated contact, eventually reaching their strength limit and resulting in delaminated regions. However, in the lubrication condition with m5W-30 at 80 °C (Figure 6d), no delaminated regions were observed. Figure 6b shows a preliminary stage before the formation of delaminated regions, displaying a deformed metal layer accompanied by a superficial crack. It is highly likely that the debris removed

through this low-cycle fatigue micro-mechanism with material detachment (Figure 6a–c) was only capable of scratching the surface (forming grooves) in the case of the tests performed with 5W-30 at 80 °C (Figure 6b).

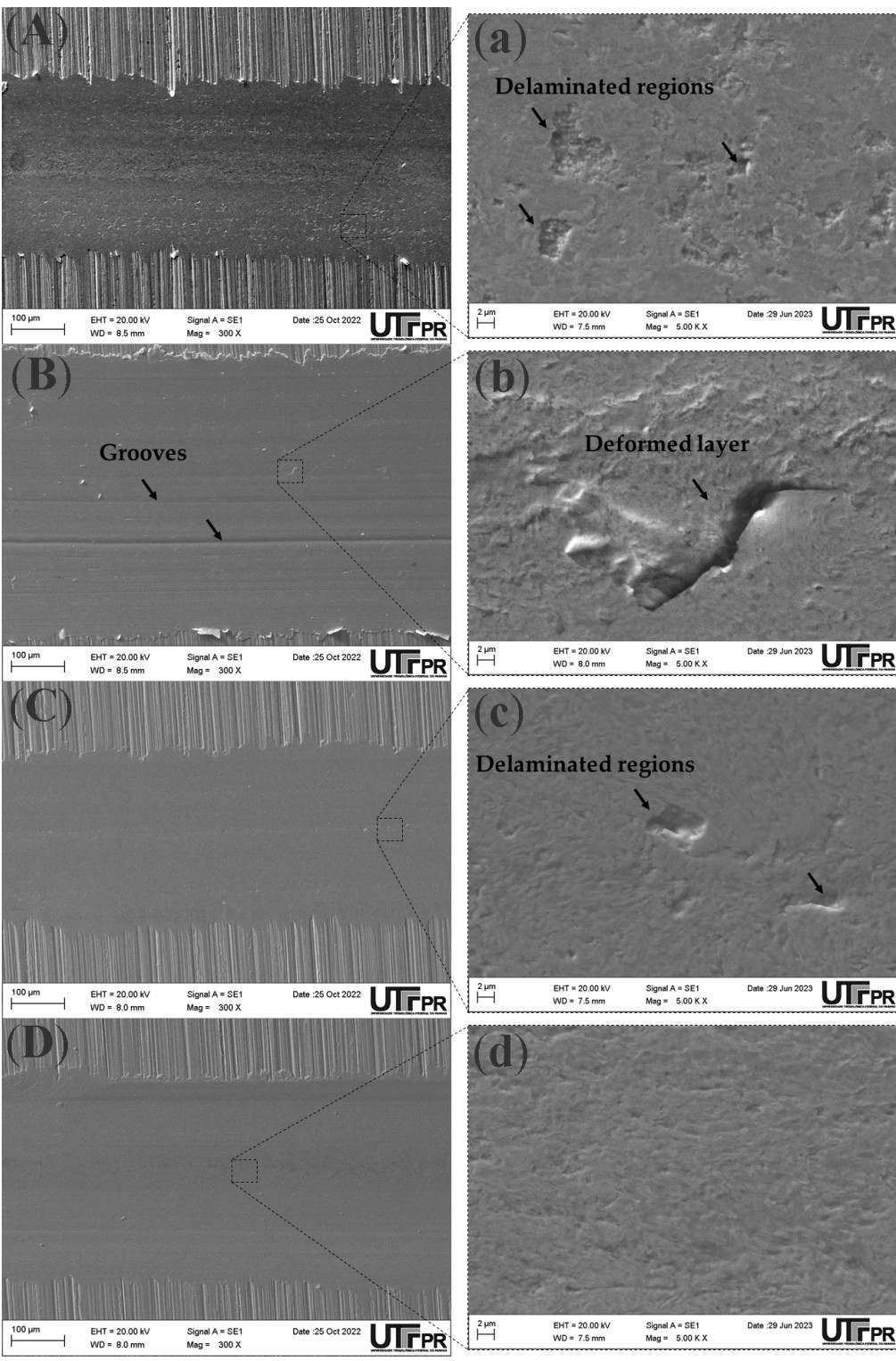

**Figure 6.** SEM micrograph of wear tracks: (**A**,**a**) SAE 5W-30 at 40 °C; (**B**,**b**) SAE 5W-30 at 80 °C; (**C**,**c**) mixture at 40 °C; (**D**,**d**) mixture at 80 °C.

**Table 4.** Average element weight percent results from EDS analysis inside the wear track. (±represents confidence intervals for $\alpha = 95\%$, min. number of measures = 16).

| Temperature | Lubricant | Average Element Composition (wt.%) | | | | |
|---|---|---|---|---|---|---|
| | | **P** | **S** | **Cl** | **Ca** | **Zn** |
| 40 °C | 5W-30 | $0.21 \pm 0.12$ | $0.67 \pm 0.20$ | 0 | $0.80 \pm 0.30$ | $0.30 \pm 0.14$ |
| | m5W-30 | 0 | $0.24 \pm 0.10$ | $0.21 \pm 0.11$ | $0.59 \pm 0.27$ | $0.05 \pm 0.08$ |
| 80 °C | 5W-30 | $0.32 \pm 0.15$ | $1.01 \pm 0.39$ | 0 | $1.38 \pm 1.48$ | $0.23 \pm 0.15$ |
| | m5W-30 | $0.27 \pm 0.15$ | $0.87 \pm 0.30$ | $0.96 \pm 0.49$ | $1.43 \pm 0.69$ | $0.19 \pm 0.12$ |

As a general trend indicated in Table 4, lower weight percentages (wt.%) of all elements were detected in the tribofilms formed at 40 °C compared to those at 80 °C. However, at 40 °C, lower adsorption of additives containing phosphorus, sulfur, and zinc was detected in the worn areas lubricated with m5W-30 compared to those lubricated with 5W-30 oil. It appears that there is a competition for adsorption on metallic surfaces between the additives of the 5W-30 and the MC, but only when lubricated at 40 °C. Finally, the specific film thickness ($\lambda$) was calculated by considering the roughness of the samples (Sq) before (initial) and after (final) the tribotests at the midstroke (maximum speed) to access the lubrication regimes. The results are presented in Table 5.

**Table 5.** Measured Sq values (root mean square height) and calculated $\lambda$ values (specific lubricant film thickness) for the surfaces before (initial) and after (final) tribological tests.

| Temperature | State | Lubricant | Sq (μm) | $\lambda$ |
|---|---|---|---|---|
| 40 °C | Initial | 5W-30 | $0.431 \pm 0.008$ | 0.064 |
| | | m5W-30 | $0.418 \pm 0.060$ | 0.066 |
| | Final | 5W-30 | $0.172 \pm 0.080$ | 0.155 |
| | | m5W-30 | $0.111 \pm 0.023$ | 0.228 |
| 80 °C | Initial | 5W-30 | $0.424 \pm 0.044$ | 0.025 |
| | | m5W-30 | $0.445 \pm 0.020$ | 0.024 |
| | Final | 5W-30 | $0.151 \pm 0.050$ | 0.068 |
| | | m5W-30 | $0.118 \pm 0.024$ | 0.085 |

Table 5 confirms that the tribotests were conducted under the boundary/mixed lubrication regime ($\lambda < 1$; note: Hansen, J. showed the lubrication regime transition from mixed to full film might occur at $\lambda$ values as low as 0.61. Thus, a transition from boundary to mixed film might occur at even lower values [36]), which is typical for various components such as gears, piston rings, and rolling bearings. Additionally, Table 5 demonstrates that at both the initial (unworn) and final (worn) conditions of the surfaces, the $\lambda$ values were approximately three times lower at 80 °C compared to 40 °C. These results align with the higher wear rates observed at higher temperatures for both lubricants (Figure 4b). During the tribological contact, the initial sanded surface undergoes plastic deformation as it transitions gradually from a more severe initial contact condition (with a very low $\lambda$) to a less severe one. This phenomenon can be primarily attributed to the running-in process, as shown in the CoF graphs (Figure 5a). The running-in process involves various mechanisms, including the redistribution and leveling of surface asperities as well as the elimination of initial roughness or surface irregularities. After the running-in phase, the rubbing of the surfaces leads to wear, smoothening of the worn surface, and the formation of a protective tribofilm in certain areas, minimizing direct metal-to-metal contact. Consequently, the final $\lambda$ values were consistently higher than the initial $\lambda$ values in all cases, a finding that is numerically confirmed by Azam et al. [32]. Notably, the final $\lambda$ value was consistently higher for the m5W-30 lubricated tests compared to 5W-30 oil, indicating the formation of a smoother surface and thus an improvement in lubrication performance.

## 4. Discussion

### 4.1. Effect of the Metal Conditioner in the Degradation and Rheological Properties

In this study, it has been demonstrated that the addition of MC does not significantly impact the rheology, thermal stability, or thermo-oxidative stability of the lubricant SAE 5W-30 API SN. However, it has been validated that MC does indeed affect the tribological performance of the lubricant. Based on the results, the addition of 5% *v/v* of metal conditioner preserves the main properties of the fully formulated lubricant while improving its tribological performance. These findings are particularly relevant for lubricated engine components that operate under mixed and fluid film lubrication regimes, such as bearings and piston rings.

Maintaining the properties of a fully formulated lubricant is crucial for achieving efficient lubrication. It plays a significant role in preventing drag losses, minimizing contact severity, and prolonging the lifespan of the lubricant. By preserving the properties of the lubricant and reducing wear, overall engine performance is enhanced [1,4,6]. Therefore, the results presented in this article provide an explanation for the observed improvements in performance seen in motorcycle engines when MC is added, as investigated by [15].

In the boundary lubrication regime investigated in this research, the primary factor influencing friction and wear performance is the formation of a tribochemical-induced film [1,5,8,33]. The properties of the 5W-30 oil, including viscosity (Figure 3b) and degradation (Figure 1), remained unchanged with the addition of MC, thus exerting minimal impact on tribological performance within this lubrication regime [4,32,33]. Therefore, the observed differences in tribological performance in this study can be attributed to the tribochemical effects of the investigated lubricants.

### 4.2. Effect of the Metal Conditioner in the Tribological Properties at 40 °C

In the tribological tests conducted at 40 °C, the addition of MC to the 5W-30 oil did not yield any benefits in terms of wear or CoF (Figures 4 and 5). Nevertheless, it was observed that the tests conducted without MC (neat 5W-30) exhibited more severe wear mechanisms and a rougher worn surface (Figure 6a, Table 5) in comparison to the track lubricated with the mixture (m5W-30, Figure 6c, Table 5). In fact, the final lambda (λ) value was only 2.4 times higher than the initial value for 5W-30, whereas for m5W-30, the final λ value was 3.5 times higher (Table 5: 40 °C).

Studies by Saini et al. [35] and Azam et al. [32] have shown that the presence of tribofilm increased the lubricant film thickness (and thus λ) and also improved the lubrication regime by facilitating the entrainment of more lubricant within the contact area. This could explain why the tribofilm formed by m5W-30 reduced the contact severity at 40 °C compared to the tribofilm formed by 5W-30 oil. However, this observation appears to be inconsistent with the chemical composition of the tribofilms analyzed by SEM-EDS (Table 4). In the case of the mixture (Table 4: m5W-30—40 °C), lower adsorption of additives containing phosphorus, sulfur, and zinc was detected compared to the 5W-30 oil. This might be related to the presence of chlorine-containing additives, which resulted in reduced adsorption of P, Zn, and S compounds. These compounds likely originate from the most common AW additive in automotive lubricants, zinc dialkyl dithiophosphate (ZDDP), which has been extensively studied in the literature [1,5,33].

According to Pham et al. [34] and Jech et al. [33], in the boundary regime, the tribofilm undergoes continuous wear and replenishment during surface rubbing, leading to a continuous competition for additive adsorption on metal surfaces. Chlorine is frequently found in EP and AW additives due to its high reactivity with metallic surfaces [2–4,30,37]. As a consequence, it appears that at 40 °C, chlorine-containing additives had an antagonistic effect, limiting the adsorption of other additives. However, overall, the tribofilm formed by m5W-30 resulted in reduced contact severity without significantly altering the wear and friction response. Similar results were obtained by Petrushina et al. [37], who demonstrated that chlorine-containing EP additives exhibit greater chemical reactivity with the Fe, Cr, and Ni elements present in steels compared to sulfur-containing EP additives.

### 4.3. Effect of the Temperature in the Tribological Properties

The tribological tests conducted at 80 °C exhibited more severe contact conditions compared to those performed at 40 °C. This distinction becomes evident when examining the λ values presented in Table 5. The initial λ values were approximately 0.025–0.024 at 80 °C, whereas they were around 0.064–0.066 at 40 °C for both lubricants. These results can be attributed to the influence of higher temperatures on various factors, including the fluid pressure of the lubricant, tribofilm pressure, and substrate contact pressure required to sustain the applied load. Consequently, an increase in the wear rate was observed as the temperature of the lubricants increased (Figure 4b).

Regarding the pressure supported by the tribofilm, Jech et al. [33] demonstrated that an increase in temperature and/or shear (load, sliding cycle frequency) results in the formation of a thicker and more extensive tribofilm by a SAE grade 5W-30 oil. However, as observed in the current study, these characteristics do not show a correlation with wear [33]. According to Jech et al. [33], the growth and increased coverage area of the tribofilm with temperature may lead to weakened bonds within the film, thereby reducing its structural integrity. Consequently, an increase in wear is observed at higher temperatures.

As a general trend observed in Table 4, lower wt.% of all elements were detected in the tribofilms formed at 40 °C compared to those at 80 °C. These results agree with the model proposed by Umer et al. [8] to explain the different tribological performances in the boundary regime of commercial fully formulated engine oil (5W-30). According to [8], all additive molecules must overcome an energy barrier to activate during two-body sliding in tribological contact due to thermal or thermomechanical energy (shear). Thus, the model proposed by Umer et al. [8] is related to the results obtained in this study (Table 4): the higher activation energy during lubrication at 80 °C compared to 40 °C facilitated the adsorption of film-forming additives in both lubricants. Additionally, studies by Lyu et al. [1], Huynh et al. [6], and Pereira et al. [7] have shown that temperature is the variable with the most significant effect on improving the tribofilm formation of commercial engine oils in tribo-tests, as it influences the composition and thickness of the tribofilms.

### 4.4. Effect of the Metal Conditioner in the Tribological Properties at 80 °C

In the case of tests conducted at 80 °C, the composition of the tribofilms between 5W-30 and the mixture did not show statistically significant differences for the elements phosphorus, sulfur, calcium, and zinc, unlike the tests lubricated at 40 °C (Table 4). However, at 80 °C, the λ value at the end of the test was approximately 3.5 times higher for the m5W-30 and 2.7 times higher for the 5W-30 compared to their initial λ values. These differences can be attributed to adding MC and its effects on tribofilm properties. Therefore, it can be inferred that chlorine-containing tribofilms are the main factor explaining the differences in wear between the two lubricants at 80 °C (Figure 4b). This indicates a beneficial synergy between the additives in the MC and in the 5W-30 oil at 80 °C. Since the contact conditions are more severe at 80 °C than 40 °C (lower λ values at 80 °C in Table 5), the role of the AW additives in 5W-30 oil and the MC additives becomes more significant, resulting in larger differences in wear resistance (Figure 4b).

The superior effectiveness of additives containing halogens in terms of wear is related to their primary mechanism of forming sacrifice films on nascent metallic surfaces, which can be regenerated as soon as they are removed [2]. This is a consequence of the severe load and speed conditions in the boundary regime and the high local temperatures in the contact asperities [2,4]. Asadauskas, Biresaw, and McClure [30] found that chlorinated paraffins were more effective in reducing wear and friction than ZDDP in soy-based and mineral-based oils. This result was related to the higher adsorption properties of the chlorine-containing EP additives compared to ZDDP in the boundary regime, as observed in our study.

Predicting the interaction between additives is a complex task. Eickworth et al. [5] reported interactions between AW and FM additives in different base oils, ranging from positive to antagonistic and intermediate synergistic responses. Likewise, Huynh et al. [6]

highlighted the negative and positive effects of adding cyclopropane carboxylic acid and Ni-nanoparticles, respectively, in reducing wear in the presence of ZDDP in steel sliding contacts. In this study, the additives in both commercial lubricants (metal conditioner and SAE 5W-30) worked together to improve the overall tribological performance at the most severe condition (80 °C) and altered the contact severity at 40 °C, thereby increasing the specific lubricant film thickness ($\lambda$). Further investigation into the mechanisms of tribofilm formation and its structure requires advanced surface analysis techniques like XPS and XANES, as demonstrated in previous studies [7,38]. This will be the focus of our future work.

## 5. Conclusions

The rheological properties (viscosity and Newtonian behavior) as well as the thermal and thermo-oxidative properties of the lubricating oil remained unchanged when a mixture of 5W-30 oil and metal conditioner in a ratio of 20:1 was used. However, the effects of using a higher proportion of the conditioner than recommended by the manufacturer ($\geq 5\%$ *v/v*) are still unknown.

It is worth noting that the infrared absorption bands typically associated with degradation in the metal conditioner actually correspond to additives containing functional groups C=O, P–O, and P=O. This characteristic should be taken into consideration when comparing the results of used oils to avoid incorrect misinterpretations regarding the degradation of the mixtures.

The mixture of synthetic oil 5W-30 with a metal conditioner in a ratio of 20:1 resulted in reduced wear at 80 °C, primarily due to the formation of a protective tribofilm facilitated by the presence of chlorine-containing additives. These additives exhibited notable anti-wear and extreme pressure properties.

At 40 °C, no significant differences in the coefficient of friction or wear rate were observed between the tests. The tribological performance at both temperatures showed beneficial synergy between the additives in the metal conditioner and the fully-formulated SAE 5W-30 oil.

**Author Contributions:** O.P.C.; Conceptualization, methodology, validation, formal analysis, investigation, writing—original draft, writing—review and editing, and project administration. Y.N.d.l.R.; Conceptualization, methodology, validation, formal analysis, investigation, writing—original draft, writing—review and editing, and project administration. V.B.-B.; funding acquisition, resources, and project administration. P.C.B.; Validation, investigation, resources, project administration, and funding acquisition. T.C.; Conceptualization, Validation, methodology, formal analysis, writing—original draft, writing—review and editing, and funding acquisition. All authors have read and agreed to the published version of the manuscript.

**Funding:** The authors O. P. Calabokis and T. Cousseau are grateful for the scholarships provided as part of the research project (ACT No. 03/2020). The author P. Borges specially acknowledges the *Conselho Nacional de Desenvolvimento Científico e Tecnológico*—Brazil (CNPq) for the research scholarship (Process 308716/2021-3). Y. Nuñez acknowledges the *Coordenação de Aperfeiçoamento de Pessoal de Nível Superior*—Brazil (CAPES) (Grant 88887.484833/2020-00). The APC was funded by the *Fundación Universitaria Los Libertadores*—Colombia (FULL) (Project No. ING-08-23).

**Data Availability Statement:** The data that support this work and its results are not available to be shared because they are under confidentiality agreements. Access to the data can be requested through an official document.

**Acknowledgments:** The authors thank the Multiuser Laboratory of Federal University of Technology—Paraná—Campus Campo Mourão (CAMULTI-CM) for the calorimetry analysis. Also, to the Multiuser Laboratory of Rheology—UTFPR (LabReo) and the Multiuser Laboratory of Chemical Analysis—UTFPR (LAMAQ) for the performed analysis. The authors thank the Materials Characterization Center—UTFPR (CMCM) for the SEM-EDS analysis. The authors express their gratitude to G. Pintaude for his valuable contribution to the discussion. This investigation is part of the research project (*Acordo de Cooperação Técnica* ACT No. 03/2020) between the *Fundação de Apoio à Educação, Pesquisa e Desenvolvimento Científico e Tecnológico da Universidade Tecnológica Federal do Paraná*

(FUNTEF-PR), Federal University of Technology—Paraná (UTFPR-CT) and a metal conditioner company.

**Conflicts of Interest:** The authors Y. Nuñez de la Rosa, V. Ballesteros-Ballesteros and Paulo César Borges declare no conflicts of interest. Author O. Palma Calabokis received a scholarship from the research Project (*Acordo de cooperação técnica No. 03/2020*) between the *Fundação de Apoio à Educação, Pesquisa e Desenvolvimento Científico e Tecnológico da Universidade Tecnológica Federal do Paraná* (FUNTEF-PR), *Universidade Tecnológica Federal do Paraná* (UTFPR-CT) and a metal conditioner company. The author T. Cousseau provided leadership of the mentioned project. The research is part of a doctoral research project that aims to study the composition, efficiency, and the mechanisms behind metal conditioners performance to suggest new formulations. All the authors attest that the results were verified and not influenced or biased by the research project.

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
