# Peer review of "Effect of a Metal Conditioner on the Physicochemical Properties and Tribological Performance of the Engine Oil SAE 5W-30 API SN"

_lubricants, doi:10.3390/lubricants11070305_

Round 1

Reviewer 1 Report

Comments on the paper titled:” Effect of a metal conditioner on the physicochemical properties 2 and tribological performance of the engine oil SAE 5W-30 API  SN

General View

The main subject matter of the paper is interesting and practical interest. However, the presentation is sometimes unclear. These are some areas where improvement is mandatory:

1. The authors note in the abstract: “The mixture improves wear resistance by around 33% when lubricated at 800C. (In relation to SAE 5W-30? Or 400C mixture?). How are all studied parameters related? Yet, “The tests lubricated with the mixture at 400C showed less severe wear mechanism” (in comparison to what?).

2. The authors study non-isothermal scanning in different atmospheres at 220- 4250C. In addition, the authors clarify that knowledge of the thermal degradation temperature is less relevant in internal combustion engines since the lubricant is in contact with oxygen and because the lubricant's life depends on its oxidation stability. Rheological measurements were performed at 200, 400 and 1000C, whereas tribological tests were performed at 400 and 800C. However, if we look at the automotive industry, the contact temperature is usually more than 800C. Therefore, 400C friction and wear tests do not correspond to real contact conditions.

3. The hardness and roughness of the steel sphere (or ball) should be presented. The temperature near contact is a critical parameter and should be measured. Noted in The Tribological Characterization the application of EDS analysis is absent. The work would be much better if the wear track profiles are presented. The authors evaluate the wear volume when the standard parameter allowing comparison of wear properties is recommended to be used: mm3/Nm.  

4. Figure 5 illustrates the coefficient of friction (COF), and the volume loss (μm³) of the moving bodies. COFs are practically the same, while wear volume depends strongly on temperature. Unfortunately, the authors do not explain why tribological films do not affect friction but vary the wear properties of the studied materials. Interestingly, the wear volume of the mixture under friction at 400C is about 3 times lower than friction at 800C. In contrast, this difference in Fig 6 is ~ 1.15 times smaller. Here, the authors suggest that “Although the wear loss was similar at 400 C, the wear tracks lubricated with neat lubricant presented adhesion, plastic deformation, and low cycle fatigue as the main wear mechanisms in SAE 5W-30 oil, while the mixture only showed plastic deformation." The authors should provide justification for considering issues. Detailed high-magnification SEM images should be described.

5. The authors should explain some contradictions in the presented version: “Higher activation energy during lubrication at 80 °C, compared to 40 °C, facilitates the adsorption of film-forming additives responsible for anti-wear action. However, anti-wear properties are better at 400C. Yet, it is noted: “Despite the differences in wear mechanisms and the composition of the tribofilms, when the tests were carried out at 400 C, the COFs and wear volume were the same for both conditions (Figure 5). However, the authors continue, it is expected that in more extended tests, 5W-30 wear volume tends to increase due to the wear mechanism associated with adhesion and low-cycle fatigue”.

Based on the analysis of the presented article it can be concluded that a major revision is needed for publication of this manuscript.

.

Comments on the paper titled:” Effect of a metal conditioner on the physicochemical properties 2 and tribological performance of the engine oil SAE 5W-30 API  SN

General View

The main subject matter of the paper is interesting and practical interest. However, the presentation is sometimes unclear. These are some areas where improvement is mandatory:

1. The authors note in the abstract: “The mixture improves wear resistance by around 33% when lubricated at 800C. (In relation to SAE 5W-30? Or 400C mixture?). How are all studied parameters related? Yet, “The tests lubricated with the mixture at 400C showed less severe wear mechanism” (in comparison to what?).

2. The authors study non-isothermal scanning in different atmospheres at 220- 4250C. In addition, the authors clarify that knowledge of the thermal degradation temperature is less relevant in internal combustion engines since the lubricant is in contact with oxygen and because the lubricant's life depends on its oxidation stability. Rheological measurements were performed at 200, 400 and 1000C, whereas tribological tests were performed at 400 and 800C. However, if we look at the automotive industry, the contact temperature is usually more than 800C. Therefore, 400C friction and wear tests do not correspond to real contact conditions.

3. The hardness and roughness of the steel sphere (or ball) should be presented. The temperature near contact is a critical parameter and should be measured. Noted in The Tribological Characterization the application of EDS analysis is absent. The work would be much better if the wear track profiles are presented. The authors evaluate the wear volume when the standard parameter allowing comparison of wear properties is recommended to be used: mm3/Nm.  

4. Figure 5 illustrates the coefficient of friction (COF), and the volume loss (μm³) of the moving bodies. COFs are practically the same, while wear volume depends strongly on temperature. Unfortunately, the authors do not explain why tribological films do not affect friction but vary the wear properties of the studied materials. Interestingly, the wear volume of the mixture under friction at 400C is about 3 times lower than friction at 800C. In contrast, this difference in Fig 6 is ~ 1.15 times smaller. Here, the authors suggest that “Although the wear loss was similar at 400 C, the wear tracks lubricated with neat lubricant presented adhesion, plastic deformation, and low cycle fatigue as the main wear mechanisms in SAE 5W-30 oil, while the mixture only showed plastic deformation." The authors should provide justification for considering issues. Detailed high-magnification SEM images should be described.

5. The authors should explain some contradictions in the presented version: “Higher activation energy during lubrication at 80 °C, compared to 40 °C, facilitates the adsorption of film-forming additives responsible for anti-wear action. However, anti-wear properties are better at 400C. Yet, it is noted: “Despite the differences in wear mechanisms and the composition of the tribofilms, when the tests were carried out at 400 C, the COFs and wear volume were the same for both conditions (Figure 5). However, the authors continue, it is expected that in more extended tests, 5W-30 wear volume tends to increase due to the wear mechanism associated with adhesion and low-cycle fatigue”.

Based on the analysis of the presented article it can be concluded that a major revision is needed for publication of this manuscript.

.

Author Response

Our responses are attached

Reviewer 2 Report

You have presented an extensive experimental study with many valuable results.

 A few remarks about it:

- Why do you use both measuring systems in the rheometer (plate-plate and couette)? Please give a short explanation.

-for the shear thinning behavior you give explanations from the literature. Do you also have your own explanations deviating from this? If yes, please specify.

- How did you measure the wear volume ? Height x width of the wear track? Please specify. Does the thickness of the tribofilm that forms play a role in this?

- Could the tribofilm be responsible for the different appearance of the wear track?

- Its a pity that you don't do any theoretical work

Author Response

Our answers are attached

Reviewer 3 Report

In general,  the research work presented doesn't provide any new scientific work; it is just a repetition of the work of earlier reported studies, as reported in [9–10]. Adding MC to well-defined oil is like doping additional additives and suggesting it improves the tribo-performance. 

Even in that case, the authors seem to be failing to achieve the desired objectives.

Moreover, the Results and Discussion section lacks in-depth analysis to support the claims made in the study.  Need further research and experimentation.

Thus, considering important aspects of scientific research, I believe the current manuscript needs to be redefined as a novelty and have clear objectives along with thorough analysis and experimentation in order to be considered for publication in any scientific journal.

  1. The authors need to consider what is scientific here since, as per the authors, "MC contains AW, FM, and 62 EP-type additives, along with common oil additives such as detergents, dispersants, anti-oxidants, and viscosity improvers [11–13].
  2. Authors should consider base oil (MO) to determine the effect of MC on tribological performance, if any. then compare it with a well-defined 

Grammatical errors throughout Manuscript

Author Response

Our answers are attached.

Round 2

Author Response

We appreciate your revision, our responses are attached.

Reviewer 3 Report

In present work, the authors have tried to explore the effects of the compatibility of the MC additive package with existing formulated oils.  Despite detailed  experimental analysis,  the authors are  advised to rewrite the  conclusions section with scientific clarity in a simplified way.

Here some suggestions that authors need to address:

The authors should include more literature on additives with synergistic effects in a lubricant package in some recent articles  such as https://pubs.acs.org/doi/10.1021/acsami.0c20759 https://doi.org/10.1016/j.triboint.2023.108332   

The authors can provide additional information or explanations in the supplementary information section about the implementation and usage of the Matlab code.

Regarding the statement on "competition for adsorption on metallic surfaces," authors can refer to well-established literature  for reference in Table 5. It would be great for the comprehension of  tribofilm formation if authors could provide data on elemental mapping in SI.

Authors are suggested to go through the manuscript in order to  check the correctness of some complex sentences, such as " A better understanding of the mechanisms 415 behind tribofilm formation and its structure demands advanced surface analysis using 416 techniques such as XPS and XANES, as studied by [7, 34], which is the scope of future 417 work. Since, as per the authors work, the MC does not contribute to any lubrication performance, there is no point in extending the studies to XPS and Xanes. Authors are advised to clearly  mention the implications of published studies  and then conclude accordingly, as per the objectives of the studies.

Overall, MC does not contribute to rheology, tribology, or oxidation stability. in 

Some sections are confusing, such as "" authors are requested to follow a logical coherent flow 

And "this characteristic is crucial for proper lubrication, as raised by Asadauskas, 278 Biresaw, and McClure [30]. The authors [30] highlighted that the rheological properties of 279 lubricant mixtures were of vital importance in the boundary lubrication regime (evaluated 280 by Four-ball and Twist Compression Tribotester) since, at the beginning of these tests, there is no time for tribochemical reactions that protect the surface. The results in Figure 282 confirm that the MC is not an additive that modifies the rheological behaviour when used in the recommended proportions (5% or 20:1).

Authors are advised to check the correctness of the whole manuscript throughout;

Almost 60 % of sentences are complex and loses logical coherence with previous sentence 

Author Response

(The authors gave the same response as above.)

Round 3

Reviewer 1 Report

The corrected version can be accepted for publication

Reviewer 3 Report

The authors have addressed the suggested comments.

The manuscript can be considered for publication.